

# Natural resources to control COVID-19: could lactoferrin amend SARS-CoV-2 infectivity?

Ehab H. Mattar[1], Fatma Elrashdy[2], Hussein A. Almehdar[1], Vladimir N. Uversky[3] and Elrashdy M. Redwan[1]

[1] Biological Science Department, Faculty of Science, King Abdulaziz University, Jeddah, Saudi Arabia
[2] Department of Endemic Medicine and Hepatogastroenterology, Cairo University, Cairo, Egypt
[3] Department of Molecular Medicine, University of South Florida, Tampa, Florida, United States

## ABSTRACT

The world population is still facing the second wave of the COVID-19 pandemic. Such a challenge requires complicated tools to control, namely vaccines, effective cures, and complementary agents. Here we present one candidate for the role of an effective cure and/or complementary agent: lactoferrin. It is the cross-talking mediator between many organs/cellular systems in the body. It serves as a physiological, immunological, and anti-microbial barrier, and acts as a regulator molecule. Furthermore, lactoferrin has receptors on most tissues cells, and is a rich source for bioactive peptides, particularly in the digestive system. In the past months, in vitro and in vivo evidence has accumulated regarding lactoferrin's ability to control SARS-CoV-2 infectivity in different indicated scenarios. Also, lactoferrin or whey milk (of human or other mammal's origin) is a cheap, easily available, and safe agent, the use of which can produce promising results. Pharmaceutical and/or food supplementary formulas of lactoferrin could be particularly effective in controlling the gastrointestinal COVID-19-associated symptoms and could limit the fecal-oral viral infection transmission, through mechanisms that mimic that of norovirus infection control by lactoferrin via induction of intestinal innate immunity. This natural avenue may be effective not only in symptomatic patients, but could also be more helpful in asymptomatic patients as a main or adjuvant treatment.

## INTRODUCTION

The year 2020 will be recorded in human history as the year of the COVID-19 pandemic, which infected and killed millions of people. The virus reached almost all countries, infected humans of all ages, and even affected many animals (*Uversky et al., 2020b*). The virus reaches almost all human organs, including the central and peripheral nervous systems (*Elrashdy, Redwan & Uversky, 2020b; Uversky et al., 2020a*), and has unexpected vertical and horizontal infectivity potential (*Elrashdy et al., 2020; Elrashdy, Redwan &*

Corresponding authors
Vladimir N. Uversky,
vuversky@usf.edu
Elrashdy M. Redwan,
lradwan@kau.edu.sa

*Uversky, 2020a*). The scientific community is still facing a major challenge to discover a preventive vaccine or standard therapeutic agent(s) for this virus.

In addition to immunoglobulins, the multifunctional protein lactoferrin (LF) is being considered as one of the mucosal surface defense molecules. It is a globular glycoprotein with 80-kDa, consisting of N- and C-terminal lobes. While the C-lobe contains more N-linked glycosylation sites, both parts of the LF are glycosylated. Each lobe of LF is able to bind a ferric ion concomitantly with a bicarbonate anion. However, being a metalloprotein, LF can bind other metals (e.g., zinc and manganese) in its two binding sites. LF is secreted by the exocrine glands and neutrophils and is present in most human secretions, such as tears, milk, vaginal mucus, seminal plasma, and saliva (*Conesa et al., 2008*; *El-Agamy, 2006*; *El-Fakharany et al., 2012*; *Legrand et al., 2008*; *Liao et al., 2012*; *Lonnerdal, 2009*). LF milk content fluctuates between mammalian species and, within a single species, depends on the lactation period.

LF is known to have many functions, ranging from immunoprotective to physiological ones, such as iron sequestration, where its main role is controlling free iron levels in external secretions and blood. It also interacts with lipopolysaccharides (LPS), peptidoglycan, ribonucleic acids, polysaccharides, and heparin, and displays pronounced antiviral and antimicrobial activity. This plethora of functions can be attributed to the fact that, structurally, LF is a hybrid protein that possesses ordered domains as well as functionally significant intrinsically disordered regions (*Berlutti et al., 2011*; *Oda et al., 2020*; *Ostan, Moraes & Schryvers, 2021*). LF and its lobes display noteworthy antimicrobial and antiviral activities against many infectants, including Gram-negative and Gram-positive bacteria and both enveloped and non-enveloped viruses, with these antimicrobial and antiviral potentials being dependent on the type of microbes and viruses (*Albar et al., 2014*; *Redwan et al., 2014b*). It is believed that most of the antimicrobial activities can be ascribed to the N-lobe. In addition, LF and its two lobes display significant anti-inflammatory, wound healing, anti-cancer, and immunomodulation activities. As an antiviral, LF acts primarily at the acute phase of infection, or even at the intracellular stage (e.g., during a hepatitis C infection) (*El-Fakharany et al., 2013*; *Liao et al., 2012*; *Redwan EL-Rashdy & Tabll, 2007*; *Redwan et al., 2014a*). LF impedes viral entry into the host cell in one of two ways: either by blocking the viral particles' corresponding cellular receptors, or by direct attachment to their proteins (*Redwan et al., 2014b*).

Lactoferrin's wide variety of functions could be attributed to its capacity for iron sequestration and/or the ability to interfere with the cellular receptors of both pathogenic microbes and their hosts (*Albar et al., 2014*; *Redwan et al., 2014b*). In this work, we discuss the potential of LF to combat COVID-19 by examining recent in vivo and in vitro experimental research. Two review articles were recently published concerning the potential effect of LF against SARS-CoV-2 (*Chang, Ng & Sun, 2020*; *Wang et al., 2020*). Although both of these reviews used previously published data on the effects of LF against SARS-CoV and other viruses, they did not consider the results of the more recent research focused on SARS-CoV-2. Furthermore, these previous articles (*Chang, Ng & Sun, 2020*; *Wang et al., 2020*) did not cover many interesting in vitro and/or in vivo experimental reports, which generated a wealth of important data on the molecular mechanisms of
LF action against SARS-CoV-2, on comparison of the human and animal LFs, and on determining whether the milk whey action against SARS-CoV-2 depends on LF alone or on other milk ingredients, such as human milk oligosaccharides (HMOs) and/or short-chain fatty acids. Our report fills these gaps and produces an overview of the field, providing answers to some fundamental questions, such as why breastfed children are mostly unaffected by COVID-19 (*Fan et al., 2020*; *Liu et al., 2021*; *Mitoulas, Schärer-Hernández & Liabat, 2020*). In addition, we enumerate the effects and the scenarios (based on viral and host factors) of how LF would work against SARS-CoV-2. Although the results of the potential LF use against COVID-19 are promising, a few limitations were recognized as well, such as the prominent anticoagulant effects of this protein and the peptides derived from it (*Shute et al., 2018*), which have also never been mentioned before. We hope that this report will expand the audience and will be of interest to readers with different backgrounds, from physicians to researchers working on biomedical and biological projects.

## SURVEY METHODOLOGY

For data collection, we conducted an electronic search to identify a set of comprehensive studies containing eligible data. To this end, Medline database through PubMed and Scopus, WEB of Knowledge, and Collabovid databases were searched to obtain related articles published from December 2019 up to January 15, 2021. The articles were scanned based on their titles and abstracts. The subject terms used in the search were human lactoferrin, bovine lactoferrin, animal lactoferrin, in vitro effects, in vivo effects. Each of these individual keywords alone or in different combinations were further pooled with "COVID-19" and/or "SARS-CoV-2" and/or "coronaviruses". Then the retrieved data were classified according to the different categories, such as review articles, research articles (published in peer-review journals or posted preprints), and clinical trials.

### Results of literature search

Through this literature search, more than 100 studies were identified that were directly or indirectly related to the description of the interplay between LF and COVID-19 and LF-SARS-CoV-2 interactions. We selected articles that were most related to COVID-19, with priority being given to the articles that reported in vivo and/or in vitro experimental data on the analysis of the LF-SARS-CoV-2 interactions that were not covered in previous reviews. Most of the experimental reports were dedicated to the in vitro evaluation of the potentials of the full-length LF of human or animal origin against coronaviruses (virulent or pseudovirus) using many cell lines culture systems. Only two reports were dedicated to the evaluation of the effects of formulated LF on COVID-19 patients, and a single study described the results of the analysis of the whey milk antiviral activity.

## DUAL LACTOFERRIN FUNCTIONS

LF is a multifunctional protein utilizing at least 8 types of receptors, such as a) lactoferrin receptor/LRP-1/CD91/apoE receptor or the chylomicron remnant receptor, b) intelectin-1

(omentin-1), c) TLR2, d) TLR4, e) CXCR4, f) CD14, g) Heparan sulfate proteoglycans (HSPGs), and h) interleukin-1 distributed in almost all cells of the human body, as reviewed in (*Kell, Heyden & Pretorius, 2020*). These different kinds of receptors enable LF to perform its various physiological and immunological functions.

It is known that LFs of different origin may show differences in their antiviral and antibacterial activities. For example, camel LF (cLF) showed higher inhibitory effects on the hepatitis C virus (HCV, genotype 4a) than human, bovine, and sheep LFs (hLF, bLF, and sLF, respectively). It is also likely that cLF might show significant anti-SARS-CoV-2 activity. There are at least three characteristics that make cLF unique among the LFs from other species (*El-Fakharany et al., 2013*): 1) the glycosylation pattern of cLF is different from those of other LFs; 2) some residues in cLF, such as $Pro_{418}$, $Leu_{423}$, $Lys_{433}$, $Pro_{592}$, $Gly_{629}$, $Lys_{637}$, $Gln_{651}$, and $Arg_{652}$ that are critical for domain movement are different from those in LFs from other species, suggesting the existence of specific differences related to structural dynamics of cLF; 3) the N- and C-lobes of cLF are characterized by different iron release mechanisms, losing their iron at acidic pH (below pH 4.0) and neutral pH 6.5, respectively. Therefore, unlike other LFs and transferrins, cLF acts as both an iron-binding protein (LF) and an iron transporter protein (transferrin) (*Khan et al., 2001*), since it loses about half of its iron content at neutral pH, and the other half is lost at acidic conditions (pH 4.0–2.0). Curiously, it was shown that the lipid peroxidation can be inhibited by treatment with bovine LF (bLF) (*El-Fakharany et al., 2017*; *Konishi et al., 2006*), whereas cLF may show a dual-action, inhibiting lipid peroxidation and regulating the hepatic-iron content due to being able to bind and transport iron at a wide pH range (acidic to basic) (*Khan et al., 2001*).

## A direct effect of lactoferrin on viral particles

As previously established (*Albar et al., 2014*; *Liao et al., 2012*; *Redwan et al., 2014b*), most of the LF forms (natural, recombinant, full length, N-lobe, or C-lobe) exert their antiviral activity through three standard strategies (Fig. 1):

1. Targeted cell protection
2. Targeted treatment of the infected cell (intracellular viral inhibition)
3. Viral neutralization (extracellular inhibition)

In vitro evaluation of the activity of (bovine/human) LF against the virulent SARS-CoV-2 using several cell-lines (Vero E6, Caco-2, adenocarcinoma human alveolar basal epithelial cells, A549, Huh7, HEK293T cell expressing ACE2 (293T-ACE2), HCT-8, Calu-3, and MRC-5 cell lines) clearly showed that LF is capable of inhibition of viral activity, in a dose-dependent manner, via the three aforementioned strategies (*Campione et al., 2020b*; *De Carvalho et al., 2020*; *Fan et al., 2020*; *Mirabelli et al., 2020*). In addition to these observations, another study emphasized that the anti-SARS-CoV-2 activity is not limited to LF alone, but can be expanded to other components found in total whey milk (*Fan et al., 2020*). Whey milk could be even more effective in blocking/neutralizing the viral attachment, entry, and post-entry replication. This study showed that whey protein

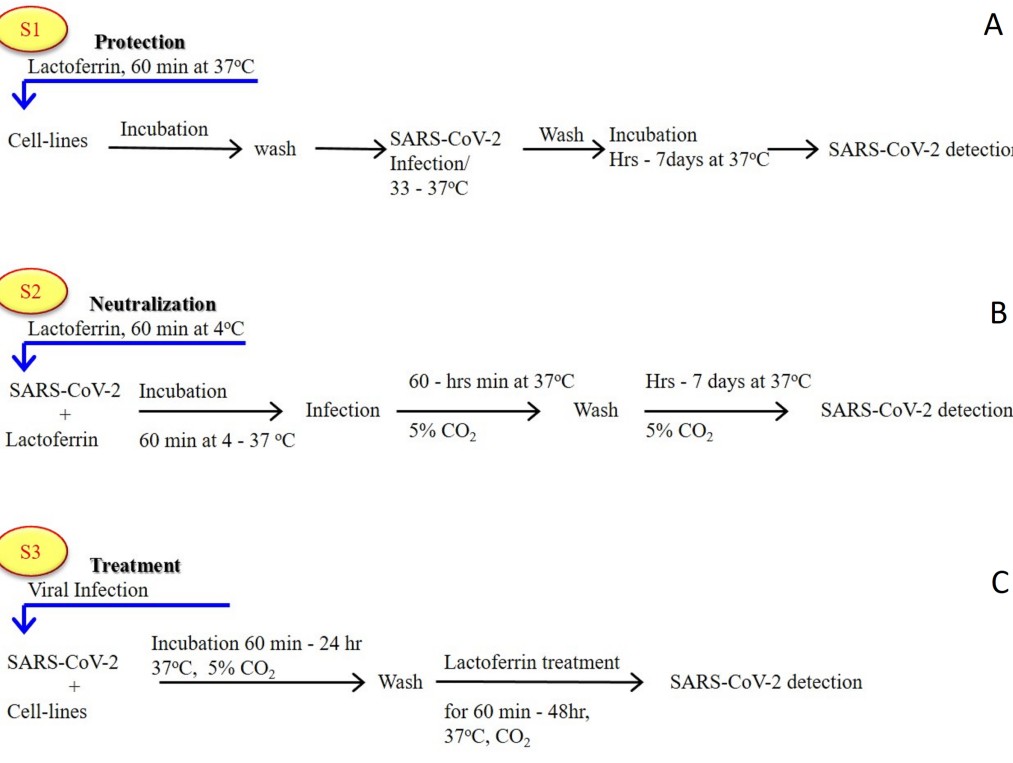

**Figure 1 General in vitro evaluation scenarios (S1–S3) of LF capacity against virus.** Each scenario has a specific incubation time. These in vitro evaluation scenarios are used in studies of HCV, SARS-CoV-2, and many other viruses. They can be used to analyze the antiviral activities over different cell lines and/or experimental conditions. (A) S1 is the protection scenario to evaluate the protective capacity of any antiviral agent. (B) S2 is the neutralizing scenario, which is applied for evaluation of the neutralizing potential of any antiviral agent. (C) S3 is the treatment scenario used in evaluation of the therapeutic capacity of any antiviral agent.

from an animal (goat and cow) inhibited the infectivity of both the SARS-CoV-2 pseudovirus and the related GX_P2V virus, though the degree of inhibition was lower than that of human whey protein (*Fan et al., 2020*). These findings suggest that human whey protein contains higher concentrations of antiviral components and/or antiviral components with higher potency than those found in other species.

Although *Fan et al. (2020)* excluded the possibility that the anti-SARS-CoV-2 activity was due to the whey milk IgA, since the human milk was collected before the emergence of SARS-CoV-2 (*Fan et al., 2020*), one could not exclude the possibility of milk fermentation during this time, which subsequently created various types of small ($\leq$3 kDa) bioactive peptides (*Madadlou, 2020*). Furthermore, free fatty acids generated by milk lipases (such as lipoprotein lipase and the bile salt-stimulate lipase) could destroy many RNA-enveloped viruses of different families (*Conzelmann et al., 2019*). Antimicrobial, anti-ACE, anti-hypotension, and anti-hypertensive inhibitor bioactive peptides were created not only from human milk but were also obtained from sheep, bovine, goat, buffalo, and camel milk (*Abd El-Salam & El-Shibiny, 2012*; *Dave et al., 2016*; *Rahimi et al., 2016*; *Rai, Sanjukta & Jeyaram, 2017*; *Wada & Lönnerdal, 2020*; *Yahya, Alhaj & Al-Khalifah, 2017*). However, IgA antibodies from recovered patients were shown to be
able to block the interaction of ACE2 and SARS-CoV-2 (*Ejemel et al., 2020*). This conclusion was supported by *Wettstein et al. (2020)* who provided evidence that alpha-1 antitrypsin (α1-AT) inhibits SARS-CoV-2 (virulent isolate and pseudoparticles) infection through targeting the viral spike protein and then blocks the SARS-CoV-2 infection of human airway epithelium at physiological concentrations (*Wettstein et al., 2020*).

In addition to human whey milk proteins, peptides, glycopeptides, and short-chain fatty acids (SCFAs), there are hundreds of bioactive structures of human milk oligosaccharides (HMOs), which are the third most abundant solid components of whey milk, after lactose and lipid. In fact, about 5–15 g/L of HMOs are present in human milk (*He, Lawlor & Newburg, 2016*; *Ramani et al., 2018*; *Triantis, Bode & Van Neerven, 2018*). As a result, HMOs are known to play fundamental roles in infant and adult bodies through three major functions related to innate immunity: 1) inhibition of binding of various pathogens to their corresponding receptors due to their structural homology with the host-mucosal cell surface receptors; 2) prebiotic activity and shaping the intestinal microbiota; 3) immunoregulation and modulation of inflammation. This unique structural repertoire of HMOs is restricted to human milk, and is very limited or entirely missing in bovine milk (*Ramani et al., 2018*), which may explain why human whey milk was more effective against the SARS-CoV-2 and GX_P2V viruses (*Fan et al., 2020*). These results may have a fundamental answer for the question of why COVID-19 morbidity and mortality in children and infants are very limited. LF was also shown to be capable of inhibiting infection of human cells by the SARS pseudovirus (*Lang et al., 2011*), and its production is up-regulated during the coronavirus infection about 150-fold (*Reghunathan et al., 2005*). In the rat model, the ingestion of exogenous LF enhanced protection against sepsis-induced acute lung injury (ALI) (*Han et al., 2020*; *Han et al., 2019*). It is also possible that the exogenous intake of LF by ingestion of milk products might enhance protection against SARS-CoV-2 infection, as shown in recent published results, which indicated the inhibitory roles of food/milk ingredients against cellular SARS-CoV-2 entry (*Fan et al., 2020*; *Hoffmann et al., 2020*; *Madadlou, 2020*; *Ou et al., 2020*; *Sano et al., 2005*). However, it should be noted that PF-4, LF, IL-8 (CXCL8), and polyarginine neutralize the anticoagulant activity of heparin (*Shute et al., 2018*), particularly in COVID-19 patients associated with thrombocytopenia and LF and IL-8 upregulation. Respiratory secretions contain high LF concentrations (0.1–1.0 mg/ml), where it completely neutralizes the heparin anticoagulation effects at 10 μg/ml (*Shute et al., 2018*). Also, LF could exert antithrombotic activity through its ability to regulate the plasminogen activation and coagulation cascade control (*Kell, Heyden & Pretorius, 2020*; *Zwirzitz et al., 2018*). This antithrombotic activity is not limited to lactoferrin, but is extended and confirmed by the ability peptide sequence derived from it (*Xu et al., 2020b*).

Recently published experimental data revealed that both the holo- and apo-forms of LF (holo-LF and apo-LF) can effectively inhibit entry and replication of SARS-CoV-2, thereby acting as effective inhibitors of SARS-CoV-2 infection with an $IC_{50}$ of 308 nM, and these protein forms were capable of potentiation of the efficacy of both remdesivir and hydroxychloroquine (*Mirabelli et al., 2020*). Furthermore, *Hu et al. (2021)* reported that bLF has synergistic antiviral effect with remdesivir on coronaviruses (*Hu et al., 2021*).

The evidence for the existence of such synergism supported the translational potential of LFs as broad-spectrum antivirals that can be used against coronaviruses, including SARS-CoV-2. Although both human and bovine LFs possessed the dose-dependent capability to inhibit viral entry into all analyzed cell types, bovine LF showed a significantly higher reactivity than human LF, with corresponding $IC_{50}$ values of 44.9 and 466 nM, respectively (*Hu et al., 2021*; *Mirabelli et al., 2020*). Although bLF was more effective than hLF, *Hu et al. (2021)*, in an in vitro comparison study that utilized many cell-line systems and a battery of LF concentrations revealed that both human and bovine LFs have a wide spectrum of inhibitory activity against many types of human coronaviruses (HCoV-229E, HCoV-NL63, and HCoV-OC43), which agrees with the results of the antiviral activities assay of bLF/hLF against a pseudovirus of SARS-CoV-2 in all cell-lines tested. These results clearly indicate that the inhibition of SARS-CoV-2 pseudovirus entry by LFs is cell type independent (*Hu et al., 2021*). Therefore, it seems that the molecular mechanisms behind the LF inhibitory effects can be attributed to the blockage of heparan sulfate proteoglycans (HSPGs) by LF (*Hu et al., 2021*).

Therefore, screens generated important results demonstrating that LF acts as an efficient SARS-CoV-2 inhibitor in vitro with multimodal efficacy. In addition, LF was able to retain its anti-SARS-CoV-2 activity 24 h post-infection, suggesting the presence of some additional mechanisms of action other than simple entry inhibition (*Mirabelli et al., 2020*). In fact, during Huh 7 cell treatment with LF, up-regulation of the expression of several interferon-stimulated genes (*MX1*, *ISG15*, *IFITM3*, and *RSAD2* that encodes radical S-adenosyl methionine domain-containing protein 2 also known as Viperin) was observed (*Mirabelli et al., 2020*). Similarly, although the addition of LF (100 μg/mL) induced a partial inhibition of SARS-CoV-2 multiplication in pre-infected Caco-2 intestinal epithelial cells, this protein significantly induced and upregulated the expression of many innate and adaptive immunity markers, such as IFNA1, IFNB1, TLR3, TLR7, IRF3, IRF7 and MAVS (*Salaris et al., 2021*). Based on this in vitro potential of bLF against SARS-CoV-2, the authors suggested that LF combined with Vitamin D will be valid adjuvant therapeutic tool for patients with COVID-19 (*Salaris et al., 2021*).

LF also stimulated an antiviral host cell response and maintained inhibitory activity in alveolar epithelial cells derived from induced pluripotent stem cells (iPSC), which act as a model for the primary site of infection. Since LF has not been shown to have adverse effects in humans, these findings suggest that this protein can be considered as a readily translatable adjunctive therapy for COVID-19 (*Costagliola et al., 2021*; *Mirabelli et al., 2020*). Furthermore, the fermentation of milk and/or LF by gut microbiota releases many active compounds (*Cockburn & Koropatkin, 2016*) that may directly interact with the viral particles and/or modulate the immune response. In a recent study (*Figueroa-Lozano et al., 2020*), scanned the effects of N-glycans derived from bovine LF on monocyte-derived dendritic cells. This study revealed that although TLR-2, TLR-5, TLR-7, and TLR-9 were not significantly altered, the different isolated N-glycan forms from bLF possessed a tight regulation of TLR-3, TLR-4, and TLR-8, as well as increased the IL-6 production (*Figueroa-Lozano et al., 2018*).

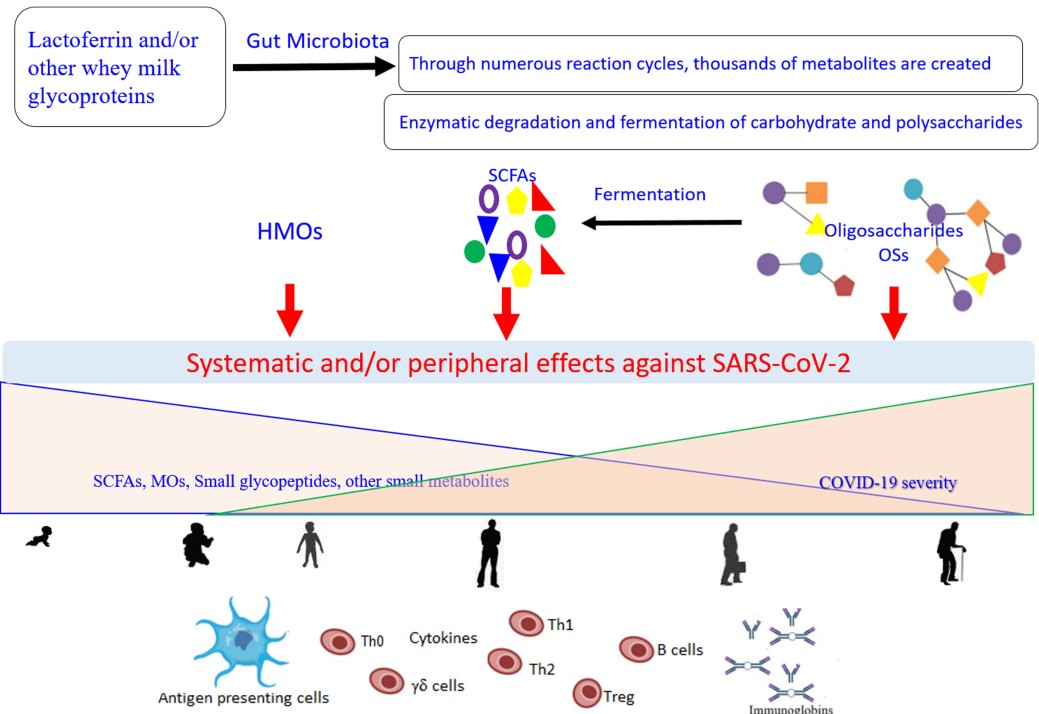

**Figure 2 General scenario for generation and effects of small metabolites created as a result of the digestion of glycosylated lactoferrin and/or other whey milk glycoproteins by different microorganisms from the intestinal microbiota, on the COVID-19 severity, aging, and their interconnection.** Metabolites can be engaged in the direct interactions with the SARS-CoV-2 particles and/or show indirect potential against the viral replication through modulation of the immune response network via the antigen presenting cells (Dendritic cell and Toll-Like receptor 2, 4, and 8). These effects dependent on the kind of food stud and gut microbiota balance, and subsequently on their concentrations and distributions that change with the age. SCFAs (short-chain fatty acids), OSs (oligosaccharides), HMOs (human milk oligosaccharides).

Of note, TLR-8 senses the viral ssRNA rich in adenylate and uridyalte, with this recognition leading to the activation of the innate immune response (*Tanji et al., 2015*). Figure 2 represents a general scenario of the creation of small metabolites via digestion of the glycosylated LF and/or other whey milk glycoproteins by different microorganisms found in the intestinal microbiota and the effects of these compounds on COVID-19 (*Cockburn & Koropatkin, 2016*; *Karl, 2021*; *Ren, Cheng & Wang, 2021*). The important roles of different type of microbiota (specifically the intestinal microorganisms) in COVID-19 development, severity, and/or recovery have been attracting the increased interest of researchers (*Costagliola et al., 2021*; *Karl, 2021*). Currently (as of February 27, 2021), there are at least 32 enrolled clinical trials using microbiota (of different source or form) in COVID-19 patients (clincaltrail.gov).

Based on the net charge, bLF has been shown to prevent viral entry into host cells utilizing competitive binding to the cell surface receptors, primarily the negatively charged compounds such as glycosaminoglycans (GAGs) (*Berlutti et al., 2011*; *Wakabayashi et al., 2014*). However, cLF is more competitive in this respect due to its highly anionic nature (*El-Fakharany et al., 2013*; *Redwan et al., 2014a*). In addition, LFs are able to inhibit

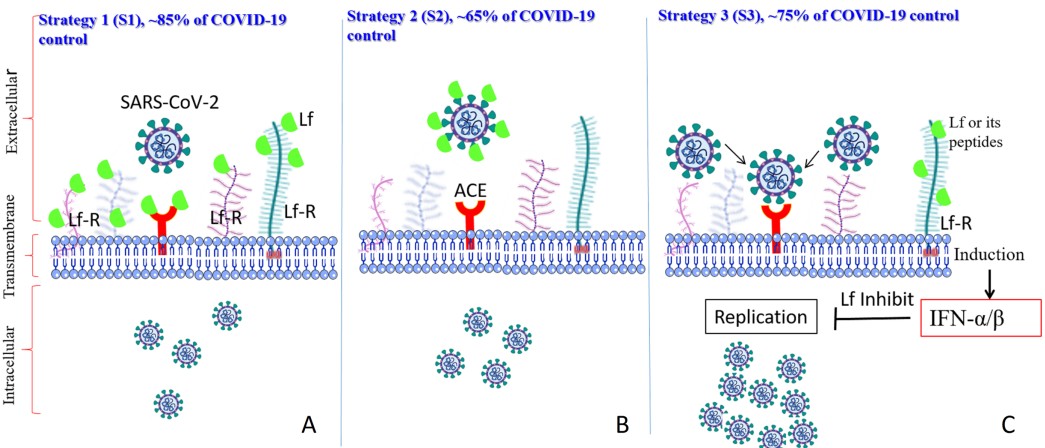

**Figure 3** Schematic representation of how SARS-CoV-2 enters the target cells, where corresponding scenarios (AS1–CS3) are based on the in vitro studies and predication. LF interacts with its corresponding host cell receptors through their glycan and/or via protein-protein interactions (AS1), LF interact with the viral spike protein to hide it from the ACE2 receptor (BS2), and/or LF interacts with its corresponding receptors, and enter the target cells, which induce many intracellular cascades inhibiting viral replication (lactoferrin is shown by a green body).

viral infections by binding to DC-SIGN and LDL receptors (*Chien et al., 2008*; *Groot et al., 2005*). Using anti-TMPRSS2 inhibitors leads to incomplete inhibition of SARS-CoV-2 entry, while complete inhibition was reached by using both anti-TMPRSS2 as well as cathepsin L/B inhibitors (*Hoffmann et al., 2020*; *Ou et al., 2020*). It is worth noting that many food bioactive peptides and proteins could inhibit SARS-CoV-2 (*Madadlou, 2020*). Lactoferrin is one of these bioactive inhibitors for SARS-CoV-2, through inhibition of cathepsin L only at an elevated affinity ($\sim10^{-7}$) (*Sano et al., 2005*). Overall, LF shows its antiviral effects at the early phase of infection, stopping viral particles from entering the host cells by either binding to the viral particles directly or by blocking cellular receptors. LF exerts its antiviral activity through direct interaction with the surface components of the viral particles (*Berlutti et al., 2011*; *Oda et al., 2020*). For example, LF was shown to bind to the E1 and E2 proteins of HCV (*Yi et al., 1997*), to the F protein of respiratory syncytial virus (RSV) (*Sano et al., 2003*), and to the gp120 protein of HIV (*Puddu et al., 1998*). Furthermore, when exerting its antiviral activity against Echovirus 6, LF interacts with both host cells and viral particles, (*Tinari et al., 2005*). As LF is one of the mucosal membrane defense proteins, in addition to its potential of host cell receptor binding it can strongly bind viral particle glycoproteins to inhibits viral entry (*Valenti & Antonini, 2005*). Lang et al., explored the role LF plays in the entry of the SARS pseudovirus into Myc cells and showed that it could block the binding of the viral spike protein to host cells, suggesting that LF performed its inhibitory function for SARS at the viral attachment stage (*Lang et al., 2011*).

The currently accepted model (Fig. 3) suggests that LF can prevent infection of the target cells by SARS-CoV-2 by interfering with the attachment factor, or by binding to host cell molecules that the virus uses as a receptor (ACE2) or co-receptors (competition) such as HS-PG and Sialoside glycosaminoglycans (SIA-PG), or by direct binding to virus

particles, as described for herpesvirus, polio, rotavirus, and R5 and X4-HIV-1, and/or via the intracellular localization involving inflammatory pathways or apoptosis (*Berlutti et al., 2011*; *Saidi et al., 2006*). If all these models were harnessed together, LF's anti-SARS-CoV-2 capacities would be considerably higher. In a recent analysis of the potential molecular mechanisms by which LF can interfere with SARS-CoV-2 cell invasion (*Miotto et al., 2020*) it was shown that several regions on the LF surface are capable of binding to the sialic acid receptors on the host cell membrane, thereby shielding the cell from virus attachment. Furthermore, although there is no significant shape complementarity between LF and ACE2, LF contains regions of high complementarity with the N- and C-terminal domains of the SARS-CoV-2 spike protein, with the most complementary region being the one in the C-terminal region, which is involved in the spike-ACE2 interaction (*Miotto et al., 2020*).

Therefore, the observed antiviral action of LF can be, at least in part, attributed to an efficient competition between ACE2 and LF for binding to the SARS-CoV-2 spike protein (*Burckhardt & Greber, 2009*; *Llorente García & Marsh, 2020*; *Miotto et al., 2020*). In light of its ability to enter the cell and pursue the nucleus, LF may also have abolished the cytokine storm cascade activation, therefore avoiding systemic complications as well as disease exacerbation.

Importantly, LF is also capable of exerting its antiviral activity when it is added at the post-infection phase, as demonstrated in HCV infection (*El-Fakharany et al., 2013*; *Liao et al., 2012*; *Redwan et al., 2014a*), in HIV infection by *Puddu et al. (1998)*, and in Rotavirus infection by *Superti et al. (1997)*. LF's efficacy during the post-infection phase suggested that it can also be useful for impacting multiple intracellular steps of virus infection (*Kell, Heyden & Pretorius, 2020*).

## Host immune system modulation action of lactoferrin

The immunomodulatory action is an important functional feature of LF, which can not only modulate the immune system through connecting both adaptive and innate immune responses but also maintain the homeostasis of the host in several human diseases, from microbial infection to cancer development (*Eipper et al., 2016*; *Habib et al., 2021*; *Uversky et al., 2017*). It has a comparable ability to lower inflammation and positively modulate the changes associated with iron metabolism, including trafficking and storage of iron, which attributes to restoring T conventional lymphocytes ($T_{conv}$) responses (*Habib et al., 2021*; *Macciò & Madeddu, 2020*).

bLF formulated in liposomes was used in two different studies on confirmed COVID-19 patients (*Campione et al., 2020b*; *Serrano et al., 2020*). Results reported by Serrano et al. showed that a liposomal bovine LF supplement containing 32 mg of LF orally administered at 4 to 6 doses/day for 10 consecutive days, accompanied by two to three daily 10 mg zinc doses, resulted in 100% recovery of 75 symptomatic SARS-CoV-2 positive patients within 4–5 days, and the same treatment at lower dose seemed to have prevented the disease in healthy contact (*Serrano et al., 2020*). Unfortunately, that study did not offer any immunological biomarkers related to the COVID-19 infection levels before and after treatment. *Campione et al., 2020b* assessed the efficacy of a liposomal formulation of

apo-bLF in 32 COVID-19 patients with mild-to-moderate disease and COVID-19 asymptomatic patients. The scheduled dose treatment of liposomal apo-bLF for oral use was 1 g per day for 30 days (10 capsules per day, each one containing 100 mg of apo-bLF in liposomes) in addition to the same formulation administered intranasally 3 times daily (nasal spray contained about 2.5 mg/mL apo-bLFin liposome). On day 0, 15, and 30 of the study, rRT-PCR was performed to detect the SARS-CoV-2 presence; the complete blood count and chemistry panel (liver and kidney function), iron panel, coagulation profile, IL-6, IL-10, TNFα, and adrenomedullin serum levels were also evaluated. On day 15, 10 patients (31.25%) were clear from the SARS-CoV-2, while on day 30, all patients showed a viral clearance. On day 15, 5 patients previously symptomatic became asymptomatic, with a total of 17 asymptomatic and 15 symptomatic patients. On day 30, the other 6 patients, previously symptomatic at the 15-day period, became asymptomatic, resulting in a total of 23 asymptomatic patients (*Campione et al., 2020b*) . Of note, these data were published as preprint during the first quarter of 2020, when many criteria concerning the COVID-19 symptoms, diagnosis methods, conversion between diseases phases, and discharge parameters were still under development. However, this report supported the suggestion that the LF could be used as an adjunctive treatment with other drugs, such as remdesivir. Another important point of study by *Campione et al., 2020b* is that the authors conducted in silico analysis and showed that the antiviral activity of LF can be related not only to the ability of this protein to bind to SARS-CoV-2 and cells, but also to the engagement of the LF in the direct interaction with viral spike S. Such high affinity binding of bLf to the Spike CDT1 domain can affect the attachment of S to the human ACE2 receptor, and thereby alter the virus entry to the host cells (*Campione et al., 2020b*).

Although many functions of LF are dependent on its ability to chelate two ferric ions as well as bind anionic moieties, the multifunctionality of this protein is not entirely determined by its chelation potential, and LF is known to act as a potent anti-inflammatory and immunomodulatory molecule (*Habib et al., 2021*). Its anti-inflammatory activity is defined by its ability to enter the host cell via the specific receptor, followed by the translocation from the cytoplasm into the host cell nucleus. Inside the cell nucleus, LF exerts one of its main functions, regulation of the expression of various genes, including the inflammatory genes. It is able to down-regulate pro-inflammatory cytokines (such as IL-6, IL-12, IL-1β, and TNF-α) while D-dimer and ferritin were significantly decreased, and potentiate the adaptive immune response in vitro and in vivo models, as well as in clinical trials (*Campione et al., 2020a*; *Campione et al., 2020b*), where, for example, noticeable up-regulation of the IL-10 in COVID-19 was observed in patients after being treated with LF (*Campione et al., 2020b*). It is well characterized that LF is a mediator for both innate and adaptive immunity which drives the changes in the expression of local and systemic signaling molecules. This governs the balance between pro-inflammatory and anti-inflammatory, or in other words humoral and cellular, immunities. Within this complicated cross-talk, lactoferrin serves to balance/rebalance more than 93 inflammation processes associated with infections (*Drago-Serrano et al., 2017*; *Legrand, 2016*).

The most interesting point in this study (*Campione et al., 2020b*) is that the platelet count was exponentially increased in the COVID-19 treated group. COVID-19 induces thrombocytopenia as SARS-CoV-2 seems to entrap megakaryocytes and block the release of platelets (*Thachil, 2020*), as detailed in (*Kell, Heyden & Pretorius, 2020*). Meanwhile, LF rebalanced the platelet count, which may enhance the COVID-19 viral clearance (*Campione et al., 2020b*), may increase the platelets production and activation via enhancing cathepsin G, and consequently promote the innate immune responses during acute inflammation through cathepsin G (*Eipper et al., 2016*). Given that COVID-19 comorbidity patients are more susceptible to secondary infections and release LPS with its complications, it is worth noting that lactoferrin has a unique competitive binding to LPS and LPS-binding (CD14) receptors, and can directly suppress cytokine production through inhibition of NF$_k$B binding to the cytokine promoter region (*Sakamoto et al., 2006*; *Yang et al., 2020*). It exerts many functionalities through different mechanisms, i.e., interaction with extracellular matrix proteins containing Arg-Gly-Asp (RGD) such as fibronectin, and blocks its interactions with its corresponding receptor (*Sakamoto et al., 2006*), which represents a key molecule associated with lung function disorder in COVID-19 patients, as well as lung fibrosis (*Xu et al., 2020a*).

Under pathological conditions, the Myeloid-derived suppressor cells (MDSCs) are activated and are implicated in the immunological regulation of several of these pathological conditions. Although MDCSs have their distinctive gene expression and biochemical characteristics, their morphological and phenotypical characteristics are similar to neutrophils and monocytes. Often, MDCSs are absent or redundant under steady-state physiological conditions (*He et al., 2018*). The body functions of patients infected with COVID-19 orchestration are disordered, and these changes exponentially increase with the patient's age (*Uversky et al., 2020a*). Overall, the cluster of gene expression matrixes in the leukocytes revealed that 12 and 5 clusters are dominated in COVID-19 patient and healthy control, respectively. The MDSCs populations are significantly predominant in the critically ill COVID-19 patients, such as band neutrophils, metamyelocytes, promyelocytes-myelocytes, monocytoid precursor, and immature monocytes. These clusters were defined by many up-regulated gene expression such as (LFT, LPR-1, LPR-2, ADAM8, CD58, CR1, FCER1G, MGAM, MMP8, MMP9, S100A8/9) linked to the SARS-CoV-2 infection consequences (*Vadillo et al., 2020*). On the other side of LF regulatory potential, LF could bind the low-density lipoprotein receptor protein 2 (LPR-2), which is expressed on many myeloid and leukocyte cells. The myeloid cells proliferate and activate upon LF binding to its LPR-2, causing NFkB induction cascades which transform the myeloid cells into myeloid-derived suppressor cells (MDSCs) (*He et al., 2018*; *Liu et al., 2019*). The MDSCs are interesting immune homoeostatic regulatory entities, which attenuate excessive inflammation through direct regulatory cytokines (reactive oxygen ROS, nitric oxide NO, arginase, prostaglandin E2, and IL-10) or indirectly via coordination of other immune surveillance cells (T regulatory lymphocyte, natural killer lymphocyte, CD4$^+$ and CD8$^+$ T lymphocytes, macrophages, and Dendritic cells). The more interesting issue is that LPR-2 receptor expression is age-dependent, decreasing

as age increases (*He et al., 2018*). Recently, it was determined that LF could clearly inhibit the TLR-4/NF-κB/TNF-α/IL-1β pathway components which were induced via oxygen and glucose deprivation and cerebral ischemia-reperfusion (*Yang et al., 2020*).

It is well-known that LF down-regulates the pro-inflammatory cytokines TNF-α, IL-6, IL-1β, IFN-α, and IL-8. On the other hand, when orally consumed by mice, it leads to enhanced production of Th1 immunity (IL-12, IL-18, and IFN-γ), which represents the main immunity against viral infection (*Wakabayashi et al., 2014*). It also improved the transcriptional regulation in whole blood, natural killer lymphocytes, the reduction of apoptosis in monocytes, macrophage population, and increased macrophage intracellular killing, as well as significantly up-regulated IL-10 in many human diseases and experimental models (*Kawakami et al., 2015*; *Patras et al., 2019*). Oral delivery of lactoferrin could significantly increase the production of IL-10, as well as enhance the IL-18 production in small intestine mucosa (*Kawakami et al., 2015*). The orally administered lactoferrin for human volunteers for 7–8 days at 100–200 mg daily improved the production of $CD^{+3}$, $CD^{+4}$, and $CD^{+8}$ lymphocytes population, and enhanced the T helper cells, T cell cytotoxic activities, natural killer cell (NK) cytotoxicity, and serum cytokine levels (*Kawakami et al., 2015*). It is worth noting that the IL-18 in intestinal mucosa will attribute to NK and T lymphocytes activation which works in conjunction with IFN-α, IFN-β, and IL-12 to enhance the production of NK, T lymphocytes, and IFN-γ (*Kawakami et al., 2015*). Many studies support the notion that oral intake of lactoferrin enhanced the intestine health of humans and experimental animal models, through immunological effects and influences on the host's microbiota hostage (*Kawakami et al., 2015*; *Patras et al., 2019*; *Suzuki et al., 2013*). This could answer the question of why children and infants have low COVID-19 morbidity and mortality, and supports the trend to orally use milk, milk products, or lactoferrin for COVID-19 control, particularly since the digestive system comes right the after respiratory system in infection. The density distribution of ACE2 on intestinal cells is higher than that on respiratory cells (*Gu, Han & Wang, 2020*).

LF gene (*LTF*, lactotransferrin) expression has been previously shown to be highly up-regulated in response to the SARS-CoV infection (*Reghunathan et al., 2005*). LF not only increases natural killer cell and neutrophil activity, but also blocks viral entry through binding to heparan sulfate proteoglycans. Furthermore, the ability of LF to retain anti-SARS-CoV-2 activity 24 hours post-infection emphasizes the complexity of its antiviral actions exceeding simple entry inhibition (*Mirabelli et al., 2020*). LF also stimulates an antiviral host cell response and retains inhibitory activity in induced pluripotent stem cells (iPSC-cell line), a model for the primary site of infection. Although a definitive and complete mechanism of anti-SARS-CoV-2 LF action was not determined, it was shown that, upon treatment with LF, the host cell undergoes significant modulation through increased expression of several interferon-stimulated genes where a dose-dependent reduction of SARS-CoV-2 replication was observed, which was consistent with the elevated mRNA levels of IFN-β and interferon-stimulated genes (*ISG15*, *MX1*, RSAD2 (Viperin), and *IFITM3*) in LF-treated Huh7 cells (*Mirabelli et al., 2020*).

Previously, LF was shown to decrease the production of IL-6 (*Cutone et al., 2014*), which is one of the key players of the "cytokine storm" produced by SARS-CoV-2 infection (*Conti et al., 2020*; *Lagunas-Rangel & Chávez-Valencia, 2020*). It was found that LF, either of bovine or human origin, retained antiviral activity in both the holo- and apo-forms, with the latter being the component of the orally available LF supplement. LF's potential is further heightened by its ability to mitigate a high multiplicity of infection (MOI, which refers to the number of virions that are added per cell during infection) during a SARS-CoV-2 infection. As ACE2 receptors are highly expressed on the gastrointestinal tissues, the orally available LF could be especially effective in resolving the gastrointestinal (GI) symptoms that are present in COVID-19 patients (*Han et al., 2020*). The mechanisms may be similar to the mechanisms by which LF reduces human norovirus infection through induction of innate immune responses (*Oda et al., 2020*). In line with these observations, it was pointed out that the *LTF* gene polymorphisms are associated with an increased susceptibility of patients to infectious diarrhea (*Mohamed et al., 2007*). If LF reduces viral load in the GI tract, it could reduce fecal-oral transmission of SARS-CoV-2 (*Gu, Han & Wang, 2020*). Furthermore, the orally consumed LF offers another interesting way of suppression of the viral infection, where the pepsin-digested LF derivative (highly positively charged loop domain) is generated, which has an antiviral activity more than 10-fold higher than those of the full-length LF or its N-lobe (*Berlutti et al., 2011*).

## CONCLUSIONS

Lactoferrin has a long history as an antiviral chelating agent. It works directly or indirectly on the viral molecules and is available at the market for several health purposes. Currently, 142 clinical trials include LF (*Campione et al., 2020a*). The observed antiviral action of LF can be, at least in part, attributed to an efficient competition between ACE2 and LF for the binding to the SARS-CoV-2 spike protein (*Miotto et al., 2020*). It seems that SARS-CoV-2 uses a dual hit strategy, with its spike protein being able to interact with sialic acid and ACE2 receptors on the cell's surface (*Milanetti et al., 2020*). This mechanism, where the first step concentrates the viral particles on the cell surface thereby facilitating the virus localization in the vicinity of the ACE2, may have explained the high infectivity potential of SARS-CoV-2 (*Elrashdy, Redwan & Uversky, 2020b*; *Uversky et al., 2020b*). This hypothesis is supported by the *Hu et al. (2021)* reporting model of heparin docking to bLF (*Hu et al., 2021*). LF binds to cell surface HSPGs, which blocks the interaction between SARS-CoV-2 and HSPGs and subsequent viral attachment to host cells. LF can successfully oppose infectivity by binding to HS-PG and SIA-PG, or by direct binding to the spike protein (see Fig. 3). Furthermore, there are also multiple mechanisms by which LF might affect the antiviral immune response. There are thousands of suggestions for combination treatments of COVID-19. Given its limited cost, lack of adverse effects, and wide availability, LF could be an effective and rapidly deployable option for both prophylaxis and the management of COVID-19 (*Mirabelli et al., 2020*).

### Funding

The Deputyship for Research & Innovation, Ministry of Education in Saudi Arabia funded this research work through the project number (838). The funders had no role in study design, data collection and analysis, decision to publish, or preparation of the manuscript.

### Grant Disclosures

The following grant information was disclosed by the authors:
The Deputyship for Research & Innovation, Ministry of Education in Saudi Arabia: 838.

### Competing Interests

Vladimir N. Uversky is an Academic Editor and Section Editor for PeerJ. Other authors declare that they have no competing interests.

### Author Contributions

- Ehab H. Mattar conceived and designed study, performed literature search, analyzed the data, prepared figures and/or tables, authored or reviewed drafts of the paper, and approved the final draft.
- Fatma Elrashdy conceived and designed study, performed literature search, analyzed the data, prepared figures and/or tables, authored or reviewed drafts of the paper, and approved the final draft.
- Hussein A. Almehdar performed literature search, authored or reviewed drafts of the paper, and approved the final draft.
- Vladimir N. Uversky conceived and designed study, performed literature search, analyzed the data, authored or reviewed drafts of the paper, and approved the final draft.
- Elrashdy M. Redwan conceived and designed study, performed literature search, analyzed the data, prepared figures and/or tables, authored or reviewed drafts of the paper, and approved the final draft.

### Data Availability

    This is a literature review, and no raw data were generated in this study.

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
