# Peer review of "Natural resources to control COVID-19: could lactoferrin amend SARS-CoV-2 infectivity?"

_PeerJ, doi:10.7717/peerj.11303_

## Round 0.1 · original submission · Major Revisions

Based on my own reading of the manuscript and comments from the reviewers, I came to the conclusion that after extensive revisions the manuscript can be reconsidered for publication. In the revised version, I would expect that you address all critical remarks of the reviewers.

Reviewer 1 ·

Basic reporting

Editing of the manuscript would improve the presentation of the material. For instance, the opening sentence of the abstract is missing a verb. There are typographical errors in many places, such as line 142, line 231, line 233, and the figure captions, among others.

Figure 2 appears to be from a different article or re-worked from a different article. For instance, it references age and aging as well as "food stud" and a "Huge Gut Microbiota" even though neither "food stud" nor "Huge Gut Microbiota" are discussed within the text of the manuscript.

The abstract positions the manuscript as taking a look at the clinical use of LF for COVID-19, but the content focuses on basic cellular mechanisms and in vitro experiments with little attention give to clinical data.

As currently written, the manuscript organization is lacking. Several paragraphs are off topic or out-of-place (lines - 155-165, 201-206, 231-232, among others). The addition of more section headings would be useful.

The conclusion section brings up new material that has not been covered in the body of the manuscript (lines 408-413).

The authors suggest that "specialists in food science and technology" (line 100) are an intended audience for this manuscript, but as written, this is not the case.

Experimental design

The authors describe their search terms and database use. However, it is stated that preprints were included, and as far as I am aware, the listed databases (pubmed, scopus, and web of knowledge) do not include preprints in their searches. Furthermore, the manuscript reviews studies of LF in the context of other coronaviruses (for example: lines 167-172ish), but the search terms do not include "coronavirus" without "COVID-19" or "SARS-CoV-2." This evidence suggests that the terms and databases used are wider than currently presented in the methods.

Validity of the findings

The authors use COVID-19 and SARS-CoV-2 seemingly interchangeably whereas COVID-19 is the disease caused by the SARS-CoV-2 virus. This should be corrected throughout.

Statements such as, "breastfed children are mostly unaffected by COVID-19" must cite the relevant literature. For instance, this statement is flawed. All children, even those not breastfed, are mostly unaffected by COVID-19.

The statement that "the digestive system comes right the after[sic] respiratory system in infection" also must include a citation to the relevant literature.

Evidence from cell culture experiments does not make for a "readily translatable adjunctive therapy" especially when no attention is given to the dose of LF used in the in vitro experiments being described. Are the doses physiologically relevant? There is no discussion of this in the review article text or with figures 1 or 3.

Additional comments

While this manuscript covers a lot of literature, the current organizational structure and missing citations as well as limited search details severely limit my enthusiasm for this review.

Reviewer 2 ·

Basic reporting

The paper of Mattar et al. highlights the recent in vitro and in vivo studies of lactoferrin against SARS-CoV-2 infection.
1. The paper lacks of several references, for example in the sentence (lines 63-66 and 77-78). In addition, some references are not recent.

2. The direct interaction between Lf and SARS-CoV-2 needs to be further discussed. In Campione et al. 2020b was reported a decrease in viral load on both Vero E6 and Caco-2 cell line after pre-incubation between Lf and the virus. In addition, in this study, molecular docking indicates a possible interaction between the human and bovine Lf and the spike protein

3. Please add and discuss the following papers: Salaris et al. 2021 doi: 10.3390/nu13020328; Hu et al. 2021 DOI: 10.1080/22221751.2021.1888660

4. Please, discuss the resolution of the symptoms of COVID-19 patients after Lf treatment compared.
5. Please, add the dose of lactoferrin used in clinical trial

In conclusion section, add a suggestion for future clinical trials about the role of Lf against SARS-CoV-2

Experimental design

The authors should explain the results of the study design. "We selected articles that were most related to COVID-19, with priority being given to the articles that reported experimental data on the analysis of the LF-COVID-19 interactions that were not covered in previous reviews." Add a section "result" where reporting the found articles in a synthetic way.

Validity of the findings

no commen

Additional comments

The paper is well-written and highlights the newest applications of Lf in SARS-CoV2 and coronavirus.

---

## Round 0.2 · accepted · Accept

You responded adequately to the reviewers' critics, which improved the manuscript. As the topic may be of interest to a wide audience, I am pleased to accept your article for publication.